# Melting Pot Contest: Charting the Future of Generalized Cooperative Intelligence

**Rakshit S Trivedi**[*]
MIT CSAIL
rstrivedi@csail.mit.edu

**Akbir Khan**
Anthropic

**Jesse Clifton**
Cooperative AI Foundation

**Lewis Hammond**
Cooperative AI Foundation

**Edgar A. Duéñez-Guzmán**
Google Deepmind

**John P Agapiou**
Google Deepmind

**Jayd Matyas**
Google Deepmind

**Sasha Vezhnevets**
Google Deepmind

**Dipam Chakraborty**
AI Crowd

**Yue Zhao**
Northwestern Polytechnical University

**Marko Tesic**
University of Cambridge

**Barna Pásztor** [†]   **Yunke Ao** [†]   **Omar G. Younis** [†]   **Jiawei Huang** [†]   **Benjamin Swain** [†]

**Haoyuan Qin** [†]   **Mian Deng** [†]   **Ziwei Deng** [†]   **Utku Erdoğanaras** [†]

**Natasha Jaques**
University of Washington

**Jakob Nicolaus Foerster**
University of Oxford

**Vincent Conitzer**
Carnegie Mellon University

**Jose Hernandez-Orallo**
Universitat Politècnica de València

**Dylan Hadfield-Menell**
MIT CSAIL

**Joel Z Leibo**[*]
Google Deepmind
jzl@google.com

## Abstract

Multi-agent AI research promises a path to develop human-like and human-compatible intelligent technologies that complement the solipsistic view of other approaches, which mostly do not consider interactions between agents. Aiming to make progress in this direction, the Melting Pot contest 2023 focused on the problem of cooperation among interacting agents and challenged researchers to push the boundaries of multi-agent reinforcement learning (MARL) for mixed-motive games. The contest leveraged the Melting Pot environment suite to rigorously evaluate how well agents can adapt their cooperative skills to interact with novel partners in unforeseen situations [1, 2]. Unlike other reinforcement learning challenges, this challenge focused on *social* rather than *environmental* generalization. In particular, a population of agents performs well in Melting Pot when its component individuals are adept at finding ways to cooperate both with others in their

---

[*]Corresponding Authors.    [†]Contest Participant Authors.

38th Conference on Neural Information Processing Systems (NeurIPS 2024) Track on Datasets and Benchmarks.

population and with strangers. Thus Melting Pot measures cooperative intelligence. The contest attracted over 600 participants across 100+ teams globally and was a success on multiple fronts: (i) it contributed to our goal of pushing the frontiers of MARL towards building more cooperatively intelligent agents, evidenced by several submissions that outperformed established baselines; (ii) it attracted a diverse range of participants, from independent researchers to industry affiliates and academic labs, both with strong background and new interest in the area alike, broadening the field's demographic and intellectual diversity; and (iii) analyzing the submitted agents provided important insights, highlighting areas for improvement in evaluating agents' cooperative intelligence. This paper summarizes the design aspects and results of the contest and explores the potential of Melting Pot as a benchmark for studying Cooperative AI. We further analyze the top solutions and conclude with a discussion on promising directions for future research.

# 1 Introduction

As AI systems become increasingly sophisticated and interconnected, it will be critical that they be competent at cooperating, both with other AI systems and with humans. These systems should be effective at cooperation not only in settings where all agents share the same goal, but also in *mixed-motive* settings where agents have different but not mutually exclusive goals, such that gains from cooperation are available but difficult to achieve due to selfishness or other obstacles. Many real-world problems stem from agents failing to resolve mixed-motive problems such as *social dilemmas,* including scenarios where overconsumption may deplete shared resources and public good provision scenarios where there is a free rider problem [3]. The class of mixed-motive problems also includes *bargaining problems*, in which players have differing preferences over Pareto-optimal agreements, and may fail to reach compromise [4].

The emerging field of *Cooperative AI* [5, 6] is concerned with building AI systems that can help humans and machines improve their joint welfare in general environments. With this ultimate goal in mind, our objective in running this contest was to promote (differential) progress on the *cooperative intelligence* of AI systems. Inspired by earlier work from Legg and Hutter [7], we considered the following informal working definition of Cooperative Intelligence:

> *An agent's ability to achieve its goals in ways that also promote social welfare, in a wide range of environments and with a wide range of other agents.*

Capabilities in multi-agent settings may be dual-use; e.g., the ability to make credible commitments to peaceful agreements might also be useful for making coercive commitments [5]. Thus we are interested in advancing capabilities that are *differentially* useful for cooperation, relative to more general capabilities, to the extent that this is possible. Our definition of cooperative intelligence accounts for and aligns with the previous extension of Legg and Hutter approach to multi-agent systems [8]: a more realistic definition of intelligence must give more weight to environments which contain other agents with social abilities—thereby motivating our choice of the Melting Pot suite.

The Melting Pot suite provides game-like simulations of different worlds which present a variety of social situations. Melting Pot focuses on ability to generalize to new players, not new physical environments. The participants were expected to train agents (in the form of algorithms) that learned to interact with the environment and with each other in a way that encouraged them to behave cooperatively while remaining not too exploitable by antisocial others. Towards the end of the contest, the participants submitted their algorithmic solutions to be evaluated on held out scenarios which were not previously seen by the agents during their training phase or their designers during the development phase (we discuss this in further detail below). Therefor Melting Pot is a test of zero-shot generalization to new social partners.

# 2 The Melting Pot Contest

While many aspects of cooperation—such as bargaining [9], understanding others' preferences [10, 11], negotiating [12], social learning [13], and ad hoc teamwork [14]—have been the subject of previous contests, a flexible and general benchmark for comparing the cooperative capabilities

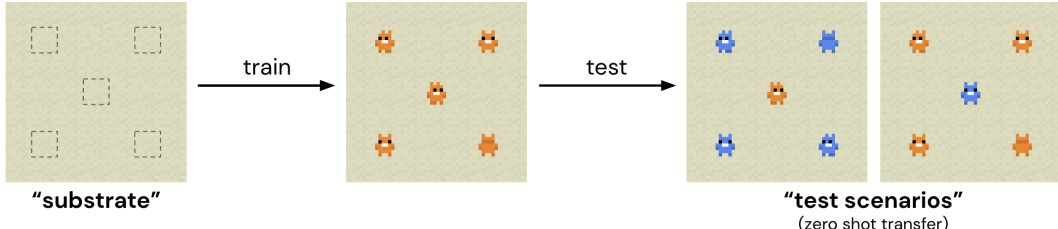

Figure 1: The Melting Pot evaluation protocol: train a focal population (orange) on a substrate, then test it in scenarios where focal agents interact with agents from the background population (blue).

of learning agents in mixed-motive settings across different environments and populations has been lacking. The Melting Pot contest directly addressed this gap and defined key metrics for cooperative intelligence to evaluate the solution approaches. This contest focused on setup (a) to test zero-shot generalization of the trained agents to new situations containing previously unseen co-players; and (b) where the number of players that will execute the trained policy at test time is unknown *a priori*[3].

Melting Pot consists of a set of test scenarios and a protocol for using them. A scenario is a multi-agent environment that tests the ability of a *focal population* of agents (to be trained by the participants of this contest) to generalize to novel social situations. Each scenario is formed by a *substrate* and a *background population*. The term 'substrate' refers to the physical part of the world: its spatial layout, where the objects are, the rules of physics, etc. The term 'background population' refers to other agents who inhabit it. While the substrate is experienced by the focal population during training, the background population is not. Figure 1 provides an overview of the Melting Pot evaluation protocol. The performance of the focal population in a test scenario measures how well its agents generalize to social situations they were not directly exposed to at training time. Therefore, the kind of generalization that Melting Pot probes is mainly along social dimensions, not physical dimensions. This is similar in spirit to other generalization-based assessments of RL (e.g., [15]), but orthogonal in its actual content since prior benchmarks mainly measure generalization along physical dimensions.

## 2.1 Substrate (Environments)

As a principled method for arriving at a subset of the Melting Pot environments to use for the contest (the full suite is described in [2]), we asked the following question: which set of environments represent a diverse and solvable challenges that require the trained agents to demonstrate capabilities for effective cooperation? To address this question, we identified four environments for this contest that vary in their complexity and that contain problems requiring various kinds of cooperative behavior to achieve a socially optimal solution. Table 1 outlines these substrates along with the list of qualitative cooperative behaviors they are expected to elicit. All Melting Pot environments are based on the DeepMind Lab2D game engine [16]. All are pixel-based ($8 \times 8 \times 3$ sprites), agents are oriented, and view the world through a partial observation window extending mostly in front of them ($88 \times 88 \times 3$).

| Substrate | Players | Potential Emergent Behaviours |
|---|---|---|
| Prisoner's Dilemma in the Matrix: Arena[4] | 8 | Reciprocity, Deception, Teaching, Sanctioning, Coalition Formation |
| Clean Up[5] | 7 | Reciprocity, Resource Sharing, Convention Following, Sanctioning, Coalition Formation |
| Allelopathic Harvest[6] | 16 | Stubbornness, Bargaining, Coalition Formation, Convention Following |
| Territory: Rooms[7] | 9 | Resource Sharing, Reciprocity Territoriality, Resisting Temptation |

Table 1: Contest substrates with a non-exhaustive list of cooperative behaviors they elicit.

---

3 This setting is important because it's common for policies that are effective at low numbers to fail when implemented by large numbers of individuals. For instance, it might not be harmful when a single agent within a large group overconsumes a shared resource but disastrous when many agents do so simultaneously.

## 2.2 Background Population

A critical component to perform zero-shot social generalization test is to evaluate the agents trained on a substrate in presence of novel co-players. This entails building intelligent background agents that can serve as useful testbed of cooperative generalization. The background agents used in the contest were either identical to or constructed using the same processes outlined in the original Melting Pot papers [1, 2]. Here, we describe the details on the construction process:

The background population consists of reinforcement learning (RL) agents, referred to as "bots" to differentiate them from the focal population's agents. The creation of the background population involved three key steps: (1) specification, (2) training, and (3) quality control.

**Specification:** The designer typically starts with an idea of what they want the final bot's behavior to look like. Since substrate events provide privileged information about other agents and the environment, we can often specify reward functions that easily induce the desired behavior. This task is much simpler than the challenge faced by focal agents, who must learn from pixels and final rewards alone. However, when a single reward function is insufficient to capture the desired behavior, we employ techniques inspired by hierarchical reinforcement learning [17, 18, 19], such as reward shaping [20] and the "option keyboard" [21].

To generate complex behaviors, we first train bots using different environment events as reward signals, similar to the approach used in Horde [18]. These behaviors are then combined using simple Python scripts, allowing us to express complex behaviors in an "if this event, then run that behavior" manner. This approach, which we call the "puppet" method, uses the same basic neural network structures (ConvNet, MLP, LSTM) as other agents but introduces a hierarchical policy structure.

For example, in the Clean Up task, we designed a bot that cleans only when other players are cleaning. The architecture is inspired by Feudal Networks [22], but with key differences. We represent goals as a one-hot vector $g$, which is embedded into a continuous representation $e(g)$. This embedding $e$ is then provided as an additional input to the LSTM. The network outputs several policies $\pi_z(a|x)$, and the final policy is a mixture $\pi(a|x) = \sum_z \alpha(e)\pi_z(a|x)$, where the mixture coefficients $\alpha(e) = \text{SoftMax}(e)$ are learned from the embedding. Notably, instead of directly associating policies with goals, we allow the embedding to learn these associations through experience.

**Training:** To train the puppet to follow goals, we train it in the respective environment with goals switching at random intervals and rewarding the agent for following them. The thing to keep in mind is that the bots must generalize to the focal population. To this end, we chose at least some bots—typically not used in the final scenario—that are likely to develop behaviors resembling that of the focal agent at test time. For instance, in Running With Scissors in the Matrix, we train rock, paper, and scissors specialist bots alongside "free" bots that experience the true substrate reward function.

**Quality control:** Bot quality control is done by running 10–30 episodes where candidate bots interact with other fixed bots. These other bots are typically a mixture of familiar and unfamiliar bots (that trained together or separately). We verify that agents trained to optimize for a certain event, indeed do. We reject agents that fail to do so.

## 2.3 Evaluation on Scenarios

When a focal population $f$ has been submitted for evaluation, the evaluation protocol measures its cooperative abilities in new social situations containing players who were never encountered during training. Each such test instance is called a *scenario*. A scenario consists of a substrate, a background population, and the number of individuals to sample from both focal and background populations.

For a given scenario, let $m$ be the number of agents sampled from the focal population and let $n$ be the number of bots sampled from the background population. Scenarios where $m > n$ are said to be in *resident mode* while scenarios where $n < m$ are said to be in *visitor mode*. For a population to achieve a high score in Melting Pot it is necessary that it score well in both resident and visitor mode scenarios. That is, the researcher designing the population must ensure that it works well despite not knowing in advance how many individuals will be sampled from it at test time. Many policies that

---

[4] For a video of *Prisoners Dilemma in the matrix: Arena*, see https://youtu.be/81QrMpsP-HU.
[5] For a video of *Clean Up*, see https://youtu.be/TqiJYxOwdxw. [6] For a video of *Allelopathic Harvest*, see https://youtu.be/BbOduMGOYF4. [7] For a video of *Territory: Rooms*, see https://youtu.be/4URkGR9iv9k.

are effective at low population densities fail to transfer to higher population density. Resident mode scenarios test the focal population's ability to absorb unfamiliar visitors and remain stable in their cooperation. Visitor mode scenarios test individuals from the focal population's ability to adapt to an unfamiliar culture. In social dilemma scenarios such as Clean Up, a more reliable solution would be for all individuals to spend some of their time acting altruistically and some of their time collecting rewards. Melting Pot penalizes overspecialization. Participants were given access to a *validation set* of scenarios from each of these classes, for each substrate. They were free to use these scenarios however they wished when designing their solutions. We created a *holdout set* of scenarios from each specified class to ensure similarity with the validation set across relevant dimensions. In the end, the participants' focal populations were scored only on the holdout set, a test of zero-shot generalization.

## 2.4 Metrics

Given a substrate $\mathcal{M}$, the per-capita return of the focal population $f$ is defined as: $\bar{R}(f \mid \mathcal{M}, \boldsymbol{c}, g) = \frac{1}{m} \sum_{i=1}^{m} c_i \mathbb{E}_{\pi_1 \sim h_1, \dots, \pi_N \sim h_N} R_i(\boldsymbol{\pi} \mid \mathcal{M})$, where $g$ is the background population of agents specific to the scenario, $\boldsymbol{c}$ is the scenario configuration represented as a binary vector indicating which $m$ players are focal, $h_i$ is the distribution over agent $i$'s policies and $R_i$ is the expected return of agent $i$ under a joint policy of all agents $\boldsymbol{\pi}$. For this contest, we consider the *(utilitarian) social welfare*[8] induced by $\boldsymbol{\pi}$ in $\mathcal{M}$ as $w(\boldsymbol{\pi} \mid \mathcal{M}) = \sum_{i=1}^{N} R_i(\boldsymbol{\pi} \mid \mathcal{M})$. This social welfare is important in defining which behaviors count as 'cooperative'. We restricted attention to scenarios in which it's possible for an agent to achieve their goals while promoting social welfare. We let $P$ be a joint distribution over scenarios $S$ that are cooperation shaping/eliciting. We defined the cooperative intelligence of $F$ as the average per-capita return it attains against this distribution, i.e., $CI(F) = \int \bar{R}(F(\mathcal{M}) \mid S) \mathrm{d}P(S)$, over scenarios that are *a priori* interesting from the perspective of Cooperative AI research.

## 2.5 Contest Structure

Thanks to our sponsors, Google Deepmind and the Cooperative AI foundation, the contest was able to award $10K in prize pool and up to $50K in compute to underrepresented groups or groups with lack of necessary compute. The contest was run in three phases: Development phase where the participant submissions were evaluated against validation scenarios which participants had access to; Practice Generalization phase where participant submissions were evaluated against a small fraction of held-out scenarios (not accessible to participants) but participants were allowed to make changes to their model based on scores they received; and Final Evaluation phase where participants selected up to three of their submissions to be evaluated against the full set of held-out scenarios and they were ranked based in the scores they received in this phase.

The unit of evaluation in Melting Pot is a *multi-agent population learning algorithm* (MAPLA). A MAPLA is a function $F$ that takes a substrate $M$ and produces a focal population, $f = F(M)$. This focal population then interacts with the background population, and obtains a particular per-capita return. The participants were required to train and submit one focal population $f$ for each of the substrates in the contest. Each participant was required to submit focal populations for each of the four substrates, and they were ranked based on their per-capita focal return.

The AICrowd platform hosted the starter kit on Gitlab. This included the baseline code provided by us and provided the submission pipeline. They also hosted the leaderboard, discussion forum, demo notebooks, an automated evaluation service, and provided teams with feedback on their performance.

## 3 Results and Analysis

The key objective of the contest was to evaluate how well different agent populations adapted to mixed-motive environments using held-out co-player populations to test *social* rather than *environmental* generalization. This section outlines the core aspects of our evaluation protocol. We begin by summarizing the results based on key metrics used to rank participants in the contest. Following this, we provide a detailed analysis of the capability profiles of agents submitted by participants. Finally, we conclude with a qualitative assessment, highlighting notable behaviors that demonstrate the cooperative intelligence exhibited by the agents in action.

---

[8] There are other appealing measures of social welfare, such as the Nash welfare or Rawlsian welfare. We use the utilitarian welfare for simplicity, in lieu of taking a stance on the best way of aggregating agents' preferences.

## 3.1 Evaluation Protocol

As described earlier, we evaluated and ranked participants' submissions based on the per capita focal return achieved by their focal agents. Each submission was assessed by collecting multiple episodes per scenario and substrate, then calculating the mean per capita focal return across these scenarios and substrates. Scores were normalized using the best and worst scores from the baseline solutions reported in the Melting Pot 2.0 paper [2]. A score above 1.0 indicates the submission outperformed the best baseline. The evaluation during the phases outlined in Section 2.5 was conducted as follows:

**(i) Development phase:** Teams were evaluated on 22 public scenarios where participants had access to the code and details on the properties of background population. A total of 73 teams submitted their agents and 8 out of 73 teams outperformed the best baseline. These evaluations were done using 4 episodes per submission and participants were allowed to make 2 submissions per day.

For the next two phases, we generated a total of 51 held-out scenarios across the 4 substrates and we used these held-out set to evaluate agents in the next two phases. The participants did not have access to the code and details on the background agents used for these evaluation phases.

**(ii) Practice Generalization phase:** Teams were evaluated on 12 held-out scenarios (3 scenarios sampled randomly from each substrate) from the full set of 51 scenarios. During this phase, which ran for 10 days, the participants received scores on the sampled 12 scenarios and they were allowed to adjust their models same as development phase based on this scores. A total of 23 teams made submission during this phase and 8 out of 23 teams outperformed the best baseline on this practice generalization test set. These evaluations were done using 20 episodes per submission.

**(iii) Evaluation phase:** Teams were not allowed to make new submissions during this phase. Instead they were required to select up to 3 submissions from development or practice generalization phase to use for final evaluation. The selected submissions were evaluated on the full-set of 51 held-out scenarios using 80 episodes per submission. For each team, we considered their best scoring submission out of the selected ones and used that score to compute final ranks. A total of 23 teams provided their selections, generating a result matrix of the size 51 x 23 focal return scores. To compute the metrics reported in following sections, we averaged these scores across scenarios and substrates. We found that 8 our of 23 teams outperformed baseline on the full held-out scenarios.

## 3.2 Summary Results

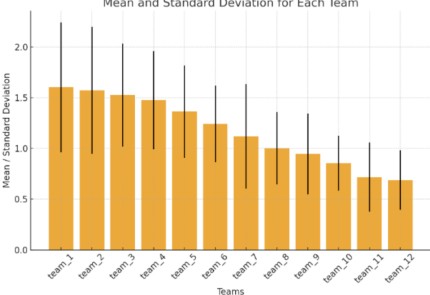
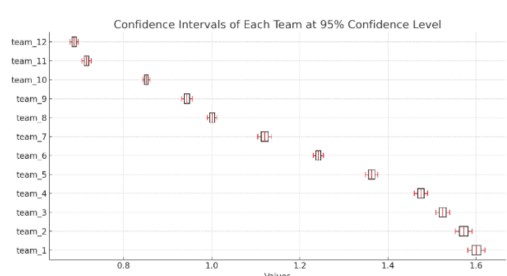

(a) Team ranking based on per capita focal return      (b) Confidence Interval on team rankings

Figure 2: Summary analysis based on Focal Returns achieved by the teams

Figure 2 provides the aggregate scores for the top 12 teams. Figure 2(a) provides the exact normalized score that was used to rank the participants. As one can observe 8 teams achieved score higher than 1.0 signifying that they performed above the best baseline. Figure 2(b) checks for overlap in the performance of these teams by considering the confidence interval at $95\%$ confidence level and we observe that there was very little overlap signifying that the team performances had significant difference between them when measured on per capita focal return. Among the teams that performed better than strongest baselines, we observed a variety of solution approaches including hard-coded and rule-based solutions, cooperative MARL solution, reward shaping and combination of imitation

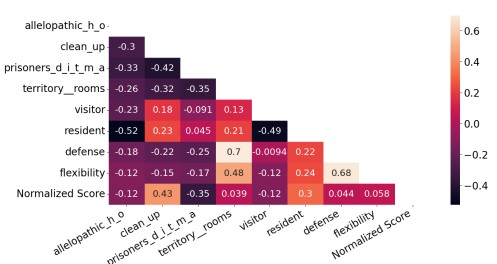

(a) Correlation matrix for the features (4 substrates, 4 tags and score).

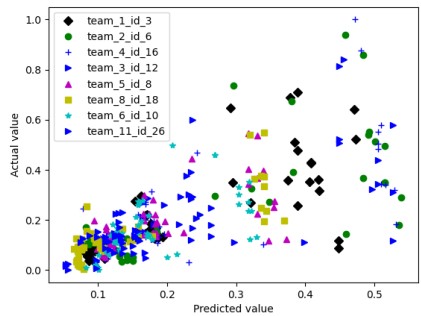

(b) Score predicted by the assessor vs actual score.

Figure 3: Feature-based analysis and predictability of the scores.

and reinforcement learning agents (refer to Section 4 for more details). Overall, the success of deep reinforcement learning approaches demonstrate a significant community-driven progress on multi-agent reinforcement learning for cooperation in presence of novel co-players.

### 3.3 Capability Profiles: A Bayesian Analysis

In addition to the aggregate performance of submission used to rank the teams for the contest, we are interested in conducting further analysis on the capabilities learned by different agents in order to support our overarching goal of informing further research in Cooperative AI. To this end, we delved into investigating the *kinds* of substrates and populations the teams are good at. In order to do this, we annotate each environment with 16 tags that describe its *cooperative demands*, including $resident$, $visitor$, $defense$, and $flexibility$, which represent whether the environment is resident or visitor (sometimes it is none of them), and whether they require defense strategies and flexibility, respectively (the other 12 tags are explained in the appendix). Figure 3a shows the correlation matrix between these features (including the substrates as features too). From these demands we can predict how well each participant would behave on new environments, provided they are annotated as well. This will also allow us to generate capability profiles for the participants.

We use two kinds of predictive models: an assessor approach [23] using a specific predictive model (XGBoost) predicting the normalized scores for all data, and a capability-based approach using measurement layouts, through hierarchical Bayesian inference [24]. For illustration, we choose eight participants (submission ids for some of the teams). For them, the assessor results show there is some degree of predictability in the score according to these features (Figure 3b).

This suggests the construction of more sophisticated, and interpretable, Bayesian models, that allow us to predict performance by comparing the demands of the environment with the capability profile of the participants. The details of the measurement layout and how good they are in predictive power are shown in the appendix, but here we include the capability profiles of these eight participants (Figure 4), because of their explanatory power. We see that the best $team\_1$ (submission $id\_3$) in aggregate score is the worst in some capabilities ($prisoner\_dilemma$, which is also observed in Figure 5b (apparently negatively correlated with score), but this participant was generally very good in the tag abilities such as $visitorAbility$, $defenseAbility$ and $flexibilityAbility$. Similarly, low-performing participants such as $team\_11$ are usually bad at all capabilities, but this one is surprisingly the best at $allelopathic\_harvest\_open$. The capability profile allows us to determine the strength and limitations of different teams in a more refined way than aggregate scores. Our benchmark has annotations for all environments, and includes the tools to build these capability profiles, helping their developers detect their strong/weak points more precisely.

### 3.4 Qualitative Analysis of Agent Behaviors

To better understand the results and assess the cooperative behavior of agents, we analyzed the trajectories of the top 12 submissions. We examined the per capita return of background agents in the presence of the participants' focal agents, the focal agents' performance per substrate, and videos of

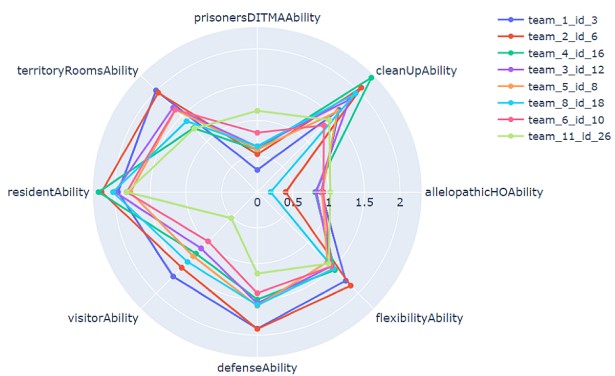

Figure 4: Teams abilities as inferred by the measurement layout

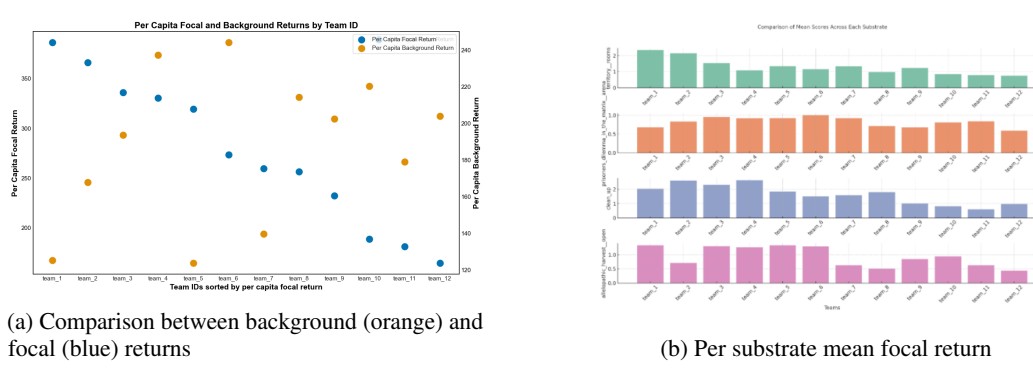

(a) Comparison between background (orange) and focal (blue) returns

(b) Per substrate mean focal return

Figure 5: Performance measures in terms of background returns and per substrate focal returns for teams ranked by the focal returns used for contest ranking

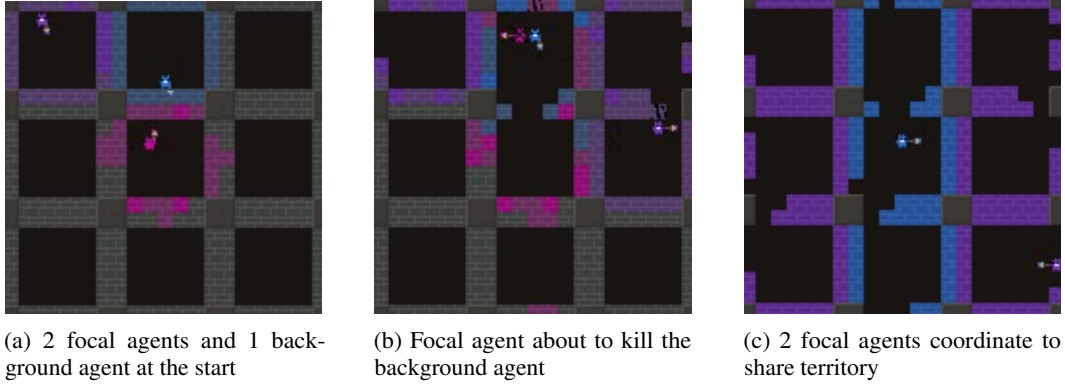

(a) 2 focal agents and 1 background agent at the start

(b) Focal agent about to kill the background agent

(c) 2 focal agents coordinate to share territory

Figure 6: Example of in-group coordination leading to anti-social behavior

the focal agents in various scenarios. Figure 5a shows that teams with high per capita focal returns often do not perform well in terms of the background population's per capita return. This indicates that while focal agents excel in the contest, they often overfit by optimizing for focal returns and within their group, failing to cooperate with background agents. This issue was particularly severe in solutions using hard-coded or rule-based policies compared to RL-based solutions.

Figure 5b reveals that higher-ranking teams did not consistently perform well across all substrates. The top two teams achieved their high ranks by excelling in a few substrates while severely underperforming in others, likely due to overfitting their agents' training to certain substrates. Additionally, videos of the submitted agents highlight in-group coordination among focal agents, often leading to anti-social behavior. Figure 6 demonstrates such a strategy: two focal agents perform in-group coordination to first kill the only background agent and then proceed to share the territory.

These observations and analyses taught us three key lessons: (i) Rule-based hardcoded policies showed more overfitting to the focal objective than RL solutions. We had assumed RL solutions wouldn't exploit the difference between focal and background agents, but we didn't account for hand-coded solutions, so teams found loopholes in that case. (ii) In hindsight, ranking submissions based on per substrate scores would have prevented teams from winning by excelling in only a few substrates. This issue might also be stemming from the final evaluation scenarios being more similar to the validation scenarios than anticipated. Designing more varied scenarios would be beneficial. (iii) Using multiple social welfare metrics for evaluating submissions would have better prevented anti-social behaviors, aligning more closely with our objective of focusing on differential progress.

## 4 Approaches submitted by participants

We outline the baseline solutions and briefly describe the best performing participant solutions (See Appendix B for more details).

### 4.1 Baselines

A baseline[9] of independent learning PPO agents implemented in RLlib was provided for participants to start building their approaches. This baseline formed the basis of the majority of submissions. Furthermore, we considered the scores from the state-of-art approaches such as Actor-Critic architecture (ACB), V-MPO [25] and OPRE [26]. A random policy and exploiter (agents trained on the test scenarios) were used for extracting the best and worst performing methods to compute the normalized score used for ranking submissions.

### 4.2 Hard-coded solution

In this approach, a set of hard-coded policies was developed, each uniquely tailored to one of four distinct substrates. Each policy was designed to strategically utilize predefined goals that were optimized for the environmental dynamics and player roles of its specific substrate. The policies included functions such as the ability to distinguish between focal and background players and mechanisms to avoid enemy zap locations. Navigation was enhanced using breadth-first search algorithms, enabling the agents to efficiently achieve their goals while avoiding obstacles. This strategic framework underpinning each hard-coded policy contributed significantly to their effectiveness, as evidenced by Figure 2a, showing that this approach achieved the highest aggregate team ranking based on score.

The high effectiveness of these hard-coded policies stemmed largely from their capacity for iterative refinement. By closely monitoring policy performance in various scenarios, developers could identify specific areas for improvement, thereby enhancing the score through targeted adjustments to the hard-coded strategies. This method proved particularly successful given the relatively intuitive nature of the game rules for each substrate, which facilitated a straightforward understanding and manipulation of policy parameters. Had the game rules been more intricate or less discernible, the scenario might have leaned more favorably towards the application of trained models, which excel in managing complex, less predictable environments.

### 4.3 State-of-art independent learning solutions

**Reward Shaping and Mixture Learning Algorithm Solution.** The reward shaping and mixture learning algorithm solution for reinforcement learning addresses key challenges with hyperparameters and reward functions, which often impede optimal performance. The implementation utilizes PopArt, IMPALA [27], and PPO [28] techniques. PopArt manages reward function scaling to prevent excessive values, PPO reduces optimization gaps with clipping techniques, and IMPALA enhances vision-based observations through scalable distributed learning. In various experimental setups, reward functions and environments were tailored to improve agent behaviors. For example, in the Allelopathic Harvest, various reward functions and training environments were tested to balance agent actions. In Clean Up, complex reward tracking promoted efficient cleaning by rewarding agents based on their actions over time. The PD in the Matrix Arena used early rewards with a specific discount factor to manage interactions effectively. In the Territory Rooms, combining team and individual play environments helped agents learn both collaborative and independent strategies.

---

[9] https://github.com/rstrivedi/Melting-Pot-Contest-2023

**Novel Agent Architecture.** This submission implemented a stable policy optimization algorithm for the agents separately, developing substrate-agnostic learning protocols, and fine-tuning the approaches to the individual challenges of the substrates. Following the baseline published in [2], it deploys Actor-Critic models such as A2C [29] and IMPALA. To generalize the learning approach across substrates and stabilize learning without tuning hyperparameters for each substrate separately, this submission implemented a PopArt [30] layer in all models and weights sharing across agents.

### 4.4 Cooperative MARL solution

This solution adopted a cooperative MARL approach to tackle the challenge of enhancing the model's generalization ability. A modified ResNet[31] was used for visual-to-state transformation, converting visual data into state information to improve state-space processing efficiency. For credit assignment, individual rewards were summed to obtain the global reward, and the Value Decomposition Network (VDN)[32] was implemented to ensure fair reward distribution among agents in multi-agent environments. In terms of cooperation and competition, the approach initially assumed cooperative behavior among players before considering their competitive behavior. However, in the Harvest and Clean environments, the absence of competitors led to some negative behaviors. Background players were added to carry out collection activities to increase player engagement. Despite achieving commendable results, the approach had limitations. The lack of directional data hindered attack strategies, incorrect color usage in visualization led to misjudging opponent actions, over-reliance on prior strategies sometimes interfered with outputs, additional ResNet layers increased memory usage, and limited inter-agent training potentially constrained performance.

## 5 Concluding Remarks and Discussions

Participants contributed a variety of solutions including state-of-the-art RL approaches based on cooperative multi-agent RL and imitation learning, as well as hard-coded approaches. Many teams were able to improve on the previously published benchmark performance metrics for Melting Pot [2] (measured on the same scenarios).

The contest has proven valuable for informing the design and structure of future contests. In this case several high performing agents focused on solving for only few of the several desired cooperative capabilities, and heavily optimized for the single evaluation metric (even at the expense of not achieving jointly prosocial behavior), overfitting too much despite the zero-shot generalization structure. We may aim to do more to discourage such strategies in future contests.

The goal is to promote *differential* progress in building agents with cooperative intelligence. The Melting Pot environment offers a wide variety of substrates with numerous capability requirements, and future contests can explore these more extensively. This contest demonstrated that Melting Pot is diverse enough that reasonable-sounding metrics do not work well across the entire suite. In the future, it would be useful to study secondary evaluation metrics like the impact on the background population, This could indicate the effects trained agents might have on humans in the real world. By analyzing the background-population per-capita return, one can determine if it is negatively impacted by the focal population. This approach helps study whether the joint policy of focal agents produces negative externalities—"side effects", impacting the broader population while sparing the focal population, aligning well with research on value alignment and AI safety.

These design insights will be directly applicable to a successor contest which will run the following year and be driven by many of the same organizers. The new contest will involve LLM-based agents, and will be based on Concordia [33].

Most notably, Melting Pot remains an unsolved problem. Results reported in the related paper and solutions received in this contest demonstrates that there is plenty of room to improve the scores in the vast majority of test scenarios and substrates. Melting Pot suite allows researchers to evaluate algorithms that train populations of agents to behave socially. How will the resulting agents resolve social dilemmas? Can they deal with freeriders? Will they learn policies that have an equitable distribution of returns? Will the winner take all? These questions not only present interesting research directions, but are also important to consider in the context of AI safety and fairness. After all, human society is a multi-agent system and any deployed AI agent will have to generalize to it.

## Acknowledgments

We thank Google Deepmind and the Cooperative AI Foundation for their generous sponsorship of this contest. We also thank AICrowd for their support in hosting the contest and providing the necessary compute resources for conducting large scale evaluations. And, we thank the anonymous reviewers for their valuable feedback towards helping us improve the quality and presentation of this report.

This work is supported in part by a Cooperative AI Foundation grant.

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

# Appendix for the Melting Pot Contest 2023 Report

## A  Further Results and Analysis

In this section, we provide extended analysis of the contest results including agent behaviors and provide more details on the capability analysis.

### A.1  Performance analysis for each substrate

Figure 7 provides a detailed view of performance of each of top 12 teams on each substrate. It further compares the difference between performance of focal agents vs the return achieved by the background agents for each substrate. This comparison lends further support to the point of focusing on different metrics for evaluating cooperative intelligence in addition to the mean focal return used in this contest.

### A.2  Assessing robustness of contest score

While our analysis has helped establish the merits considering various evaluation metrics on the top of focal per capita return, it is known that 'mean' is an aggregation technique often dominated by outlier performance. Following [34], in this section, we assess the validity of team rankings when measured with more robust metrics. Specifically, we consider **Inter-Quartile Mean:** IQM discards the bottom and top 25% of the runs and calculates the mean score of the remaining 50% runs. In our case, we consider IQM over episodes. Figure 8 shows that the team rankings under mean focal return holds when measured using IQM (although they are much closer under IQM), thereby lending confidence in robustness of overall rankings of the contest.

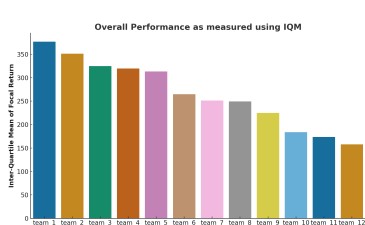

Figure 8: Overall Performance as measured using IQM

### A.3  Another example of in-group coordination: CleanUp

In section 3.3, we performed qualitative analysis of agent behaviors in a case where agents learned to perform in-group coordination leading to anti-social outcomes. In this section, we analyze another such use case, where agents again perform in-group coordination but with an intention to analyze the behavior of other agents and respond accordingly (not leading to an anti-social outcome necessarily). Specifically, in Figure 9, the scenario is such that background agents would reciprocate by cleaning the river if one focal agent starts to clean the river. In this case, the focal agents form a line at different places in the map to recognize in-group agents and then they use this information to initiate cleaning of river by background agent while switching to always consuming apples themselves.

### A.4  Details on capability profiles, measurement layouts and their predictive power

To build the evaluation models, we use the experimental data containing nine features: submission ID, four substrates ($territory\_rooms$, $allelopathic\_harvest\_open$, $clean\_up$, and $prisoners\_dilemma\_in\_the\_matrix\_arena$) and four tags ($resident$, $visitor$, $defense$, and $flexibility$). We also use the output feature, the 'Normalized Score'. To help with the interpretation of results and the output distribution of the measurement layout, we min-max scale these scores in the dataset between 0 and 1.

As described in the main paper, the first model we fit is an assessor, an XGBoost regression model from the XGBoost python library, considering 51 cases per submission (aggregated for all episodes). Fig. 10 shows the feature importance given from the library. We find that $visitor$, $clean\_up$ and $territory\_rooms$ as the features with highest importance.

The measurement layouts are built independently for each submission. So we do not use the submission Id as in the assessor model. The measurement layout basically uses these nine features, now considered as task demands, and performance of agents to infer values for each corresponding

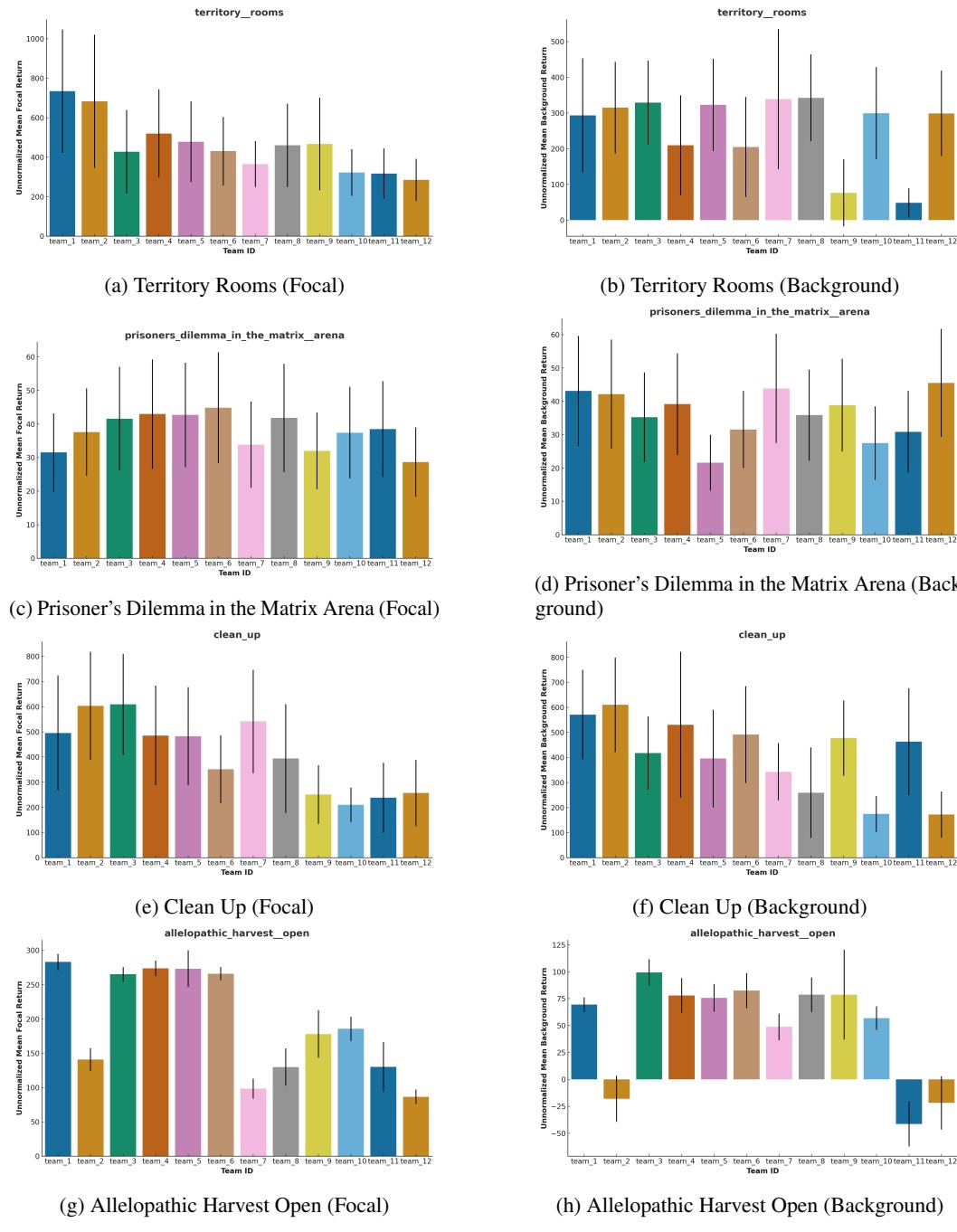

(a) Territory Rooms (Focal)

(b) Territory Rooms (Background)

(c) Prisoner's Dilemma in the Matrix Arena (Focal)

(d) Prisoner's Dilemma in the Matrix Arena (Background)

(e) Clean Up (Focal)

(f) Clean Up (Background)

(g) Allelopathic Harvest Open (Focal)

(h) Allelopathic Harvest Open (Background)

Figure 7: Comparison of Mean Focal and Background Return for different substrates

capability. See Figure 11. For more information about how to interpreting these hierarchical Bayesian networks, we refer the reader to [24].

The measurement layout in the figure shows eight abilities: allelopathicHarvestOpenAbility, cleanUpAbility, prisonersDilemmaInTheMatrixArenaAbility, territoryRoomsAbility, residentAbility, visitorAbility, defenseAbility and flexibilityAbility.

Each ability corresponds to a demand. Abilities are compared against demands, their difference is then pass through a logistic function. The outputs of the logistic functions are multiplied together with a noise variable, to model the observed performance.

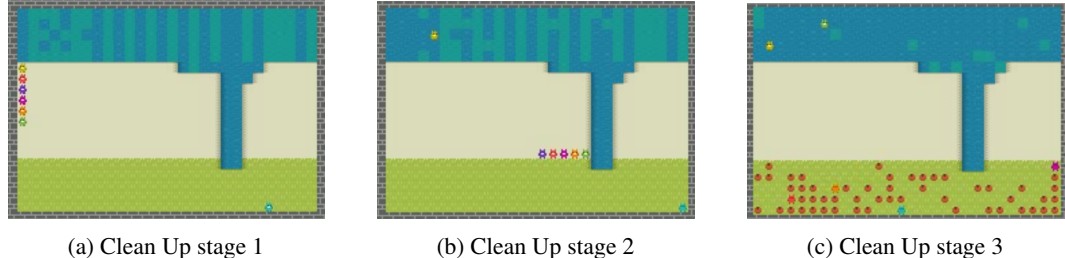

| (a) Clean Up stage 1 | (b) Clean Up stage 2 | (c) Clean Up stage 3 |

Figure 9: Progression of Agent behavior in Clean Up game

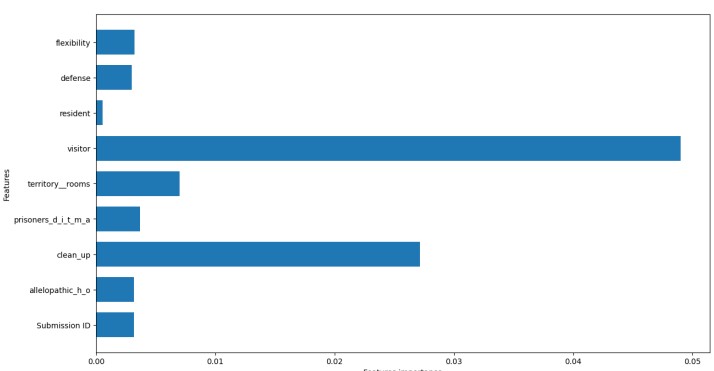

Figure 10: The average feature importance with the assessor and 408 rows.

We divide the dataset into $90\%$ training data and $10\%$ test data and repeat the experiment three times, calculating the means, seen in Fig. 4. We see that prisonersDilemmaInTheMatrixArenaAbility and allelopathicHarvestOpenAbility have a low value for all submissions. Individual team values on cleanUpAbility vary significantly, with $team2\_id5$ showing only 1.28 and $team4\_id16$ showing the highest at 2.22. On both residentsAbility, the values are relatively higher.

Finally, to validate the measurement layouts, we can look at their predictive power in terms of how well they predict performance for each submission and case in the test set. We use mean squared error (MSE) as metric. We compare the results of the assessor, the measurement layout and the aggregate mean (AggMean). AggMean is the aggregate performance for each team, i.e. a simple the success ratio on the training set. This is the common approach used for evaluating AI/ML systems (calculating and extrapolating test performance ood). The measurement layouts predict using the top-down inference of the constructed cognitive model (the graph seen in Fig. 11). The low values of MSE for the assessor support the inferred capabilities, and are much better than AggMean in predictive power (AggMean has perfect calibration but no refinement), but worse than the assessor. The assessor model, which employs XGBoost to predict performance, is not interpretable, and does not give us capability profile.

## B Substrates, Scenarios and Evaluation

In this section, we outline the details of the substrates, scenarios and evaluation setup used for the contest.

### B.1 Substrate Wrappers

As a part of our starter-kit, we provided baseline implementation of agents in RLLib for particpants to use as a base for their submissions. In order to make the melting pot environment compatible

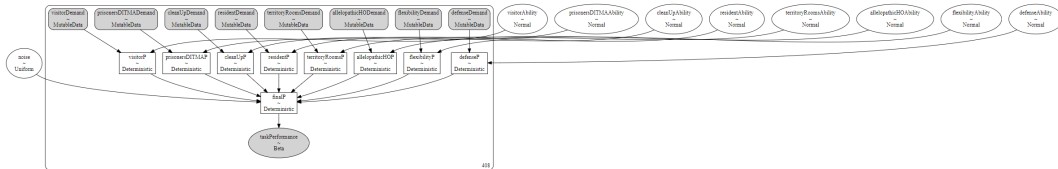

Figure 11: The measurement layout using nine features. The plate says 408 examples, but this is run separately for each of the 8 submissions, so only 51 examples are considered in each plate.

Table 2: The MSE for AggMean, Measurement Layout and XGBoost assessor.

|  | AggMean | Layout | Assessor |
|---|---|---|---|
| $team\_1\_id\_3$ | 0.048 | 0.032 | 0.014 |
| $team\_2\_id\_6$ | 0.070 | 0.040 | 0.025 |
| $team\_4\_id\_16$ | 0.071 | 0.037 | 0.026 |
| $team\_3\_id\_12$ | 0.063 | 0.045 | 0.014 |
| $team\_5\_id\_8$ | 0.024 | 0.014 | 0.011 |
| $team\_8\_id\_18$ | 0.027 | 0.014 | 0.003 |
| $team\_6\_id\_10$ | 0.017 | 0.017 | 0.007 |
| $team\_11\_id\_26$ | 0.120 | 0.012 | 0.005 |
| average | 0.042 | 0.025 | 0.012 |

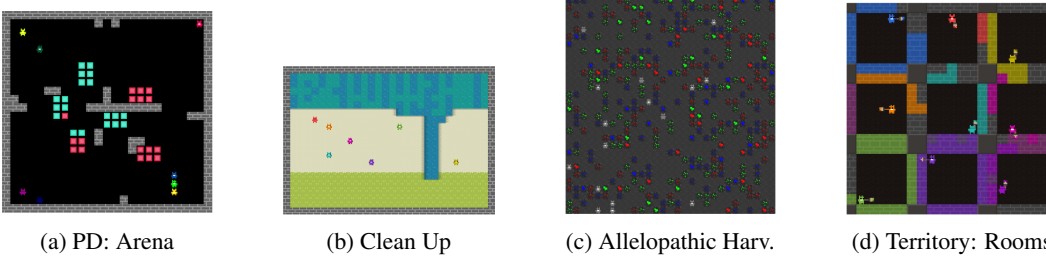

(a) PD: Arena     (b) Clean Up     (c) Allelopathic Harv.     (d) Territory: Rooms

Figure 12: Substrates considered in the Melting Pot Contest

and extensible with Rllib, we provided an interfacing wrapper[10] that provide a mapping from any MeltingPot substrate to Rllib MultiAgentEnv class and implements the necessary step() and reset() functions. Next, as noted in Section 2.1, all substrates are pixel-based and have sprite of size $8 \times 8 \times 3$ with an observation window of $88 \times 88 \times 3$. Figure 12 provides pictorial overview of the 4 substrates considered in the contest. However, during our experiments, we found that this led to slower training time on the selected environments. To address this, we bench marked the results after reducing the sprite size by a factor of 8 i.e. reducing the observation window to $11 \times 11 \times 3$. We did not find a significant difference in the performance after downsampling the sprite size and hence we provided a wrapper to downsample the sprite as a part of our starter kit. While this downsampling is done at the substrate level, the evaluation on scenario is done on contest servers (described below), where the environment always returns the observations with the full sprite size. To address this, we further provide a downsampling wrapper as a part of agent's policy such that the agent submitted by the participants always receives the downsampled sprite. These adaptations enabled support for effectively and efficiently training agents in low compute regimes, thereby attracting submissions across different academic labs and independent researchers and increasing overall participation.

## B.2 Details on Scenarios

Since Melting Pot seeks to isolate social environment generalization, it assumes that agents are familiar with the substrate. They have unlimited access to it during training. The evaluation scheme concentrates on assessing the ability of agents (and populations composed thereof) to cope with the

---

[10] https://github.com/rstrivedi/Melting-Pot-Contest-2023/blob/main/baselines/wrappers/

presence of unfamiliar individuals in the background population (in a zero-shot way), typically mixed in with familiar individuals from the population under test. A scenario is defined as the combination of substrate and background population of agents. When evaluating on scenario, the focal agents are sampled from the ones submitted by the participants while the background agents are the ones pre-trained using puppet based training pipeline described in the original paper. Scenarios are broadly categorized across two modes: If the mixture contains more focal individuals than background individuals, it is called a *resident* mode scenario. Whereas, if the mixture contains more background individuals than focal individuals, it is called a *visitor* mode scenario. As described in Section 2.4, we provided participants with access to a total of 22 scenarios across the 4 substrates during the development phase. And then we had generated 51 new scenarios that were kept as held-out set for final evaluation. These held0out scenarios were from the same distribution as development scenarios but had variations in some aspects (e.g. visitor vs resident modes, no. of focal vs background agents, reciprocating agents vs convention following agents or a mixture of both etc.). While all the scenarios (including the ones used as held-out set in the contest) will be made available publicly on the base repository[11], we exemplify the scenarios by outlining the ones available during development phase in Table 3.

Table 3: Scenarios used for Development Phase evaluation in Melting Pot Contest 2023

| Substrate | Scenario | Description |
|---|---|---|
| Allelopathic Harvest | SC 0 | Visiting a population where planting green berries is the prevailing convention |
| | SC 1 | Visiting a population where planting red berries is the prevailing convention |
| | SC 2 | Focals are resident and visited by bots who plant either red or green |
| Clean Up | SC 0 | Visiting an altruistic population |
| | SC 1 | Focals are resident and visitors ride free |
| | SC 2 | Visiting a turn-taking population that cleans first |
| | SC 3 | Visiting a turn-taking population that eats first |
| | SC 4 | Focals are visited by one reciprocator |
| | SC 5 | Focals are visited by two suspicious reciproca- tors |
| | SC 6 | Focals are visited by one suspicious recipro- cator |
| | SC 7 | Focals visit resident group of suspicious re- ciprocators |
| | SC 8 | Focals are visited by one nice reciprocator |
| PD in the matrix: Arena | SC 0 | Visiting unconditional cooperators |
| | SC 1 | Focals are resident and visited by an unconditional cooperator |
| | SC 2 | Focals are resident and visitors defect unconditionally |
| | SC 3 | Visiting a population of hair-trigger grim reciprocator bots (a) |
| | SC 4 | Visiting a population of two-strikes grim reciprocator bots (b) |
| | SC 5 | Visiting a mixed population of k-strikes grim reciprocator bots (c) |
| Territory: Rooms | SC 0 | Focals are resident and visited by an aggres- sor |
| | SC 1 | Visiting a population of aggressors |
| | SC 2 | Focals are resident, visited by a bot that does nothing |
| | SC 3 | Focals visit a resident population that does nothing |

(a) Bots who initially cooperate but, if defected on once, will retaliate by defecting in all future interactions
(b) Bots who initially cooperate but, if defected on twice, will retaliate by defecting in all future interactions
(c) Bots with k values from 1 to 3, they initially cooperate but, if defected on k times, they retaliate in all future interactions

## B.3 AICrowd Evaluation Setup

The participants were required to train the focal agents locally with no restrictions on the training approach. For evaluation, AICrowd services were used where the evaluators were hosted on multiple compute nodes. Specifically, the participants were required to submit a population of $n$ agents (where $n = 1$ was allowed) using the code provided in the baseline repository[12] and AICrowd submission

---

[11] https://github.com/google-deepmind/meltingpot
[12] https://github.com/rstrivedi/Melting-Pot-Contest-2023

kit[13]. Once the agent(s) were submitted, for each episode, the evaluator would sample focal agents from the submitted population as needed by a given scenario. During the development phase, each participant was allowed to make 2 submissions per day (with an allowance of 5 failed submissions). The participant's submitted code was packaged into a Docker image for every submission, and run on AWS. Each submission was provided with 1 vCPU and 3 GB RAM per focal agent. The resources were typically by soft-constrained using the Ray framework, while running all the agents on a single machine with 8 or 16 vCPUs, depending on the scenario. During development phase, for each scenario, the population needed to complete 4 episodes in 15 minutes. No inter-agent communication is allowed (apart from using the actions to implicitly communicate in the observation space). For practice generalization phase, the evaluation was done for 20 episodes. And for the final evaluation phase, the evaluation was conducted for 80 episodes to account for the variance in the submissions.

Further, AICrowd hosted the leaderboard to show the scores at different granularity levels to the participants. During development phase, the score were computed through evaluation on public scenarios which participants also had accessed to. During practice generalization phase (which ran for 10 days), participants received a score computed on a small sample of held-out scenarios which participants did not have access to. But the participants were still allowed to make new submissions byt adjusting their models based on the score they received. At the end of practice generalization phase, the participants were required to select upto three submission id's on AICrowd platform for the submissions they made either during development or practice generalization phase. The final scores were computed by evaluating on these selected submissions and teams were ranked according to the best score achieved.

## C   Details on Contest Submissions

In this section, we provide details on the approaches submitted by the participants of the top 6 teams in the contest.

### C.1   Team `Marlshmallows`

The submitted approach was a set of hard-coded policies designed for each individual substrate. The code[14] for these policies is available online, and the logic for each substrate is described below.

#### C.1.1   Allelopathic Harvest

The highest priority for focal agents is to navigate to the nearest ripe berry. If there are no ripe berries, the focal agents will change the color of the nearest non-red berries to red. If a green player happens to be within zap range while navigating, the focal agent will zap them. Focal agents avoid the zap ranges of background players during navigation. Two versions of this policy were submitted for the competition: one where focal agents prefer red and one where focal agents prefer green berries.

#### C.1.2   Clean Up

Focal agents initially navigate to the west wall and align side by side. At timestep 35, a role allocation occurs: the two focal agents closest to the water take on the role of dirt cleaners, while the others become apple harvesters. The dirt cleaners are programmed with specific conditions for task switching. Under one condition, they switch from cleaning dirt to harvesting apples; under another, they revert from apple harvesting back to dirt cleaning. In contrast, the focal agents designated as apple harvesters at the wall are assigned a single task; they exclusively harvest apples without any role change. To switch the goal from cleaning dirt to harvesting apples, the dirt cleaners must either see no dirt for the past 20 timesteps while in water or see at least $X$ other players in the water in the past 20 timesteps. Dirt cleaner 1 has $X$ set to 2, and dirt cleaner 2 has $X$ set to 3. Any focal agent who didn't reach the wall by timestep 35 become additional dirt cleaners with $X$ set to 2. To switch the goal from harvesting apples back to cleaning dirt, the focal agent must see no apples for the past 3 timesteps while in grass. Focal agents avoid stepping into the zap ranges of background players.

---

[13] https://gitlab.aicrowd.com/aicrowd/challenges/meltingpot-2023/meltingpot-2023-starter-kit
[14] https://github.com/benjamin-swain/meltingpot-2023-solution

### C.1.3 Prisoners Dilemma in the Matrix

Focal agents begin by navigating to resources if they are visible, or away from corners if no resources are visible. Focal agents prefer red but will navigate to green resources if there are no red visible. Once focal agents have collected at least 2 red or 2 green resources, they will navigate and interact with the nearest interactable player. If there are no interactable players visible at this point, the focal agent will make a full rotation to assist with spotting interactable players. If there are still no interactable players visible, the focal agent will continue to collect resources (or rotate in place if no resources are visible) until an interactable player is found.

### C.1.4 Territory: Rooms

The focal agents begin by running through a predefined sequence of actions (rotate and fire claiming beam). This allows the focal agents to detect which neighboring players are focal or background players; if the neighboring player does not perform the predefined action, it can be assumed to be a background player. Next, the focal agent detects the nearest reachable unclaimed or background-claimed wall and navigates to it; this continues until a background player enters or until timestep 70 when the focal agents begin to destroy walls to reach unclaimed or background-claimed walls in other areas. The focal agents make use of breadth-first search to determine the shortest path to goals while avoiding obstacles like walls and other players. The areas that the background players can zap are considered as additional obstacles. Focal agents also detect when the background players have last zapped to take advantage of the delay before they can zap again. When background players are reachable, the focal agents attempt to navigate to get the background player within zap range while avoiding obstacles. When the focal agent is injured or is in the delay period just after zapping, the focal agent will navigate to the farthest reachable area from all visible background players. Focal agents can typically only see 4 out of the 8 other players at the start of the game, so an additional method was added for focal agents to signal to each other that they are focal as opposed to background players. When two players enter each other's view for the first time, they must fire claiming beam to signal that they are focal. Additionally, if any player is seen firing claiming beam more than 8 times, that player is known to be a background player because claiming beam is only used by focal agents to signal to newly seen players, and there are only 8 other players.

## C.2 Team rookie

### C.2.1 Approach Overview

Each agent's policy is implemented as a concatenation of a ResNet module[31], a GRU layer, and an MLP layer. The MLP layer outputs the Q value of each action. We use a pre-defined prior strategy for each society game to select the best action for each agent.

Regarding mixed motivation considerations[2], we assume each agent will engage in cooperative behavior and then consider their competitive behavior. For cooperative behavior, adopt the popular Centralized Training with Decentralized Execution (CTDE) approach and use the VDN value factorization method[32].

In the **Harvest** and **Clean** environments, competitive behaviors could lead to negative effects because there are no competitors at all. To address this problem, we added some NPC agents during training; these NPC perform some collection activities, which lead to improved performance of the focused agents. Figure 13 shows our overall framework diagram.

### C.2.2 Prior Strategies

**Allelopathic Harvest** A challenging problem in this environment is how to distinguish the focal players from the background players. Since being eliminated will bring huge penalties, the cost of accidentally injuring a friend is very high. However, if you do not attack the marked player, you may be killed by the enemy. So, we considered an interesting strategy. When the berries are uniform in color, the focal player changes its color to an unpopular blue color, thereby distinguishing the focal players from the background players. However, the red and blue serial numbers were reversed during the visualization process. We were not able to discover this problem until the last day of the competition, so we achieved quite poor performance in this environment.

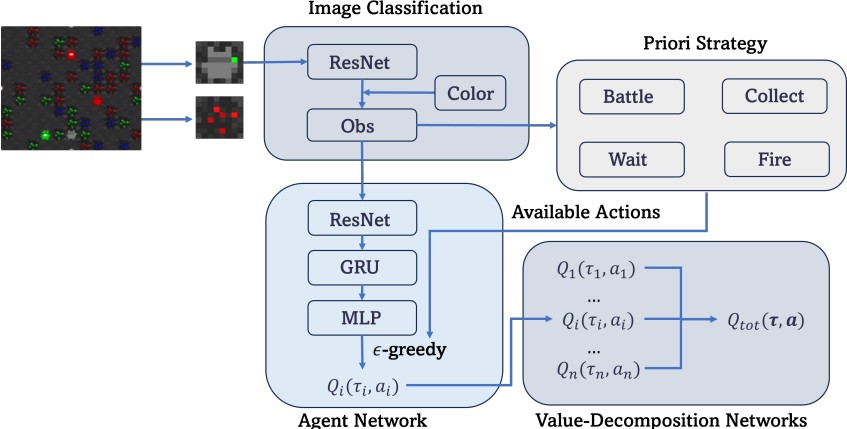

Figure 13: Architecture

**Clean Up** This is an environment that requires a choice between cleaning and eating. In theory, it is an optimal choice that two players clean the river at the same time. However, due to the presence of some selfish background players, we have to shuttle between the river and the orchard. We give certain penalties to players who stay in open spaces, empty orchards, and clean rivers and add a selfish background player during training to motivate the player's behavior. In this environment, we basically did not adopt any prior strategies.

**Prisoners Dilemma in the Matrix** This is an environment that is difficult for us to solve. The focus of this environment is still how to distinguish the focal players from the background players. We want the focal players to cooperate with other focal players but weakly cooperate or even strongly defect with other background players. Due to the existence of K-strike explosive background players, the environment encourages us to engage in defection. However, defection among the focal players will bring relatively large losses. Therefore, in this environment, we want the player's strategy to be as pure as possible. When each player is born, its strategy is defined as either cooperation or defection, and it will only collect resources that are the same as its strategy. In order to prevent the focal players from defecting each other, we set the focal players that choose the defect strategy to be unable to actively interact.

**Territory: Rooms** In our opinion, the players in this environment should be extremely aggressive. Since players cannot be resurrected, one player should be able to occupy a large amount of resources. However, for the focal players, attacking each other can take a huge toll. So, we found a way to differentiate the focal players from the background players. In the beginning, players only color the walls in the middle of the four directions. During this process, we can identify focal players who behave similarly to observations. After completing this operation, we can proceed with the normal activities, trying to encroach on the territory of other players.

### C.2.3 Limitations

The neglect of the orientation problem led to limitations in the attack strategy, and we had to rely on prior knowledge to guide the behavior of the agents. In addition, the incorrect use of colors in the visualization configurations led to a waste of time and resources in the Harvest environment, as we could not accurately identify and respond to the opponent's actions. The excessive use of prior strategies sometimes interferes with the output of the GRU model, so we need to utilize 2 to 3 consecutive frames of input to accurately predict the behavior of the adversary. At the same time, we realize that the increase in the number of ResNet layers will take up too many memory resources, so we need to carefully calculate the resource requirements when designing the model. In the training, we did not train against agents, which may also be one of the reasons for the final performance limitation. However, in the environment where competition and cooperation coexist, the task of effectively training agents becomes a complex challenge, and we can only rely on a prior strategies to distinguish between focal players and background players.

### C.3 Team `Tess-v1`

#### C.3.1 Reward shaping solution

In this approach, agents are observed periodically, and the reward model parameters are adjusted according to the difference between the goal and the learned actions.

**Implementation**

First, the approach uses a PopArt implementation for all the experiments and IMPALA model for visual observation. For policy optimization, it uses PPO with clipping technique with minibatches but it hasn't used multiple epochs almost for all training in order to achieve small steps between updates. For all the substrates, the reward functions were between the range -0.1 and 1. Specifically for prisoners dilemma in the matrix arena substrate, we have an additional fully connected layer in order to feed with the inventory informations of the agents. For all the experiments, discounted return was the same as IMPALA implementation without off-policy correction, because it has been trained purely on-policy method. In particular, we can write n-step return as, $G(t) = R_{t+1} + \gamma R_{t+2}... + \gamma^n \upsilon(S_{(t+n)})$ where $\gamma$ is the discount factor, $R$ is the reward which is returned by environment, and $\upsilon$ is the discounted value approximation which is returned by the model.

Below we outline the substrate specific reward functions:

**Allelopathic Harvest Open**

Reward function 1:

- +1 for eating any color of berry; +1 for replanting red; +1 for killing an agent which plants any other color than red
- -1 for replanting any other color
- -0.1 for killing an agent which plants red

Reward Function 2

- +1.25 for eating any color of berry
- +0.5 for killing an agent which plants any other color than red
- The other values are same as function 1

The reason why we used two different functions for this substrate is that after training with function 1, we observed the agent gives more value to zap the enemy agents and it switched to learn reward function 2 for the rest of the training. The training has been made with 8 focal agents and 8 random policy agents. We noticed that self-play training is not enough for this kind of reward functions. Because after some training, nobody plants any color other than red and the agent forgets the value of zapping the enemy agents. In the second training, we changed the environment to 12 random policy agents and 4 focal agents in order to be sure 0.5 reward is valuable for the agents. This way it gets lower reward but with higher possibility to get. We used the same agent for green lover agents because the team couldn't find a good way to train two different agents who can collaborate with each other. Instead, we created a really dominant red lover agent. Also for the PopArt implementation, it used three value heads; eating rewards, replanting rewards and zapping rewards.

**Clean Up**

Reward function:

- +0.1 for eating an apple
- +0.1 for cleaning the lake

For the clean-up substrate, the reward function is a little bit complex than the others. The environment holds the cleaner's data for the last 10 time steps and whenever any agent eats an apple, the cleaners get rewards based on their data. This way cleaners also learn how to clean for increasing the apple production possibility. Spamming the clean action is not enough, it should also learn where to clean. For the PopArt, it has two value heads; eating apple and cleaning the lake.

**Prisoners Dilemma In The Matrix Arena**

Reward Function:

- +1 for collecting cooperate box
- +30 for interaction with cooperate
- -10 for destroying cooperate task
- -1 for collecting defeat box

We used different gamma values, 0.95 for this substrate. The agents can't play the game for a while after they've been in an interaction with each other. It causes them to not get future rewards which means early rewards are always much more important than the future rewards.

**Territory Rooms**

Reward Function:

- +0.75 for capturing a wall
- +1 for killing an agent
- -0.05 for destroying a wall

For the territory rooms substrate, we used two different type of environment. The first one is created for team play, where nine focal agents playing with each other. And the second one is created to improve individual play for the exploration of the map, where one agent basically does what he wants to do while the others don't do anything. We used 4 parallel environments and 3 of them were type 2 environment and 1 of them was type 1 environment. This way it can learn capturing the whole map with individual playing and also it learns to be a team player in type 1. Also for environment type 1, it doesn't get any reward for capturing a wall if it is captured by an agent before.

### C.4 Team `monkey_king`

This solution builds a rule-based policy for each substrate. The process is divided into two parts: (i) Extract the attributes of each sprite from the RGB observations, that is, parse the specific semantics for each sprite; and (ii) Build a rule-based policy based on the extracted semantics.

**Extract the attributes from RGB observations**

We parse the attributes of one sprite by identifying the RGB value at some specific positions in the 88 × 88 observation without downsampling. The attributes include background type, whether there is an agent, the direction of the agent, the color of the agent, etc. There may be different attribute types for different substrates, and the attributes of each sprite are represented by a dictionary.

**Build substrate-specific rule based policies**

In this phase, detailed rule-based policies are designed to be employed for each substrate. While deferring the readers to Appendix for full details, we briefly provide an example of usce rules for each substrate.

In **Allelopathic Harvest**, the agent first checks whether it is currently in the cooldown period for zapping, whether there are green or blue agents that can be zapped, and whether there are no red or gray agents within the beaming range. If these conditions are met, the zap action is executed. If the zap action is not performed, the agent then considers whether there are non-red unripe berries within the planting range. If so, it will plant red berries. If neither of the above actions are performed, the agent will move towards ripe berries. The agent evaluates the four directions based on its role and the current observation, and selects the action with the highest score. The scoring criterion involves multiplying the reward of the ripe berries by a distance-discounted weight to obtain the score of the berry, and then adding the scores of the berries in each direction to determine the score of that direction. In the scenario where there are no ripe berries in the observation, the agent will move to the nearest unripe berries. Note that even if the agent's role is to prefer green berries, it will still plant

red berries. However, green berries will be given a higher score than red berries when choosing the moving directions.

In **Clean Up**, if *Timesteps*< 30, the focal agents first determine their orientation by turning viewpoint based on initialized position, then plan a path based on direction and location to meet the river that runs from the reservoir above to the grass below, forming a line in the order in which they arrive and remaining still. After testing, they can almost successfully reach the designated position in less than 30 steps. *Timesteps = 30.* Since all focal agents using the same strategy have reached the designated rendezvous point and formed a certain order, they begin to assign roles according to the number of their teammates. The allocation scheme is as follows:

- Focal agents number = 2: one "conditionally clean" and another "eat".

- Focal agents number = 3: one "clean" and two "eat".

- Focal agents number $\geq$ 4: one "clean", one "conditionally clean", the rest "eat".

If an agent on its way to a meeting point is hit by a background agent using a beam and moves out of the game and fails to reach the meeting point within 30 moves to identify a teammate, it is assigned the "conditionally clean" role by default. *Timesteps > 30.* Perform tasks according to their assigned roles. The role of "clean" stays in the sewage pool to clean up, the role of "eat" stays in the apple orchard to eat apples, and the role of "conditionally clean" changes its role according to its observation for a period of time. If there are few pollutants in the field of vision in the previous period of time, it will eat apples. On the other hand, if the number of apples in view is very low a while ago, it will clean up. In *Prisoners Dilemma in the matrix: Arena*, the agent upon rebirth, initially orients itself in a reasonable direction and then proceed towards the center. Upon detecting a desired resource within its observation, the agent will employ depth-first search to identify the shortest obstacle-avoiding path. If it encounters another agent heading towards the same resource, the agent will switch to an alternative resource within its observation. Once a sufficient quantity of resources (set to 5 in our final version) has been collected, the agent will initiate active interaction. It will continue moving towards resources to identify potential interactive partners, as the likelihood of encountering other agents near resources is typically high. After discovering another interactive agent within a certain range of vision, the agent will move towards it. While the agent actively seeks cooperation, it will shift its approach to defection (set to 5 in our final version) if it experiences repeated defections. The agent will collect red resources and interact with other agents thereafter.

In **Territory: Rooms**, if *Timesteps* < 35, the strategy is first to go around agent's own room and paint all the walls of the room in their own color, without using a claiming beam to paint the opposite room.

*Timesteps in [35, 100).* The agent stands in the center of the room and constantly looks around to monitor. If it observes someone using the claiming beam to paint the walls of their room another color, it will retaliate by using the claiming beam to paint both its own and the other person's walls their own color. If the agent observes someone trying to break through a wall to invade its territory, it will go to "high alert", which means walking up to the broken wall and waiting, and once the opponent breaks through the wall and enters his territory, the agent uses the zapping beam to remove opponent from the game. If the opponent does not use the zapping beam again to destroy the wall and enter after waiting for a period of time, the agent will return to the center point and continue to look around the monitoring.

*Timesteps in [100, 200).* Add some aggressiveness to the agent. If the agent looks around and sees that a wall adjacent to his room is unoccupied, the agent will use the claiming beam to capture it and then return to the center of the room to look around. Surveillance also involves defending against other agents trying to invade their territory.

*Timesteps $\geq$ 200.* The aggressiveness of the agent is further strengthened. In the game, standing in the center of the room can see most of the room above itself, so if the agent finds that most of the walls in the room above have not been claimed, the agent will break the walls to invade the room and attack the people in the room. After a successful intrusion, the agent will paint all the surrounding areas of the room in its own color, then return to the center of the room, look around, and if there is a room that satisfies the same condition, it will invade the next time.

### C.5 Team MeltingTeam

Our general approach for this competition was to implement stable policy optimization algorithms for the individual agents, develop substrate-agnostic learning protocols, and fine-tune the approaches to the individual challenges of the substrates. During the competition, each of us took ownership of one substrate while sharing insights and takeaways with each other. We had a weekly meeting to discuss progress, agree on the next tasks, and divide the workload.

**Substrate Agnostic Implementation**

We primarily followed the MeltionPot 2.0 whitepaper [2] for the agent's policy optimization, however, we deviated from it at certain points based on our experiences or implementation difficulties. Our codebase is built on the baseline code provided by the organization. We tested the following approaches for the agents' individual policy optimization algorithms.

1. We evaluated PPO [28], A3C, A2C, [29] and Impala [27] implementations in RLlib and decided to work mainly with A2C and Impala based on performance and stability.

2. We augmented A2C and Impala with a PopArt layer [30] that we found valuable to generalize parameters across substrates. It also facilitated a faster and more stable convergence, especially, for the substrates in which rewards could grow rapidly and change orders of magnitudes (e.g. Territory Rooms). We found that lowering the learning rate of the PopArt layer is important to avoid local optima.

3. We implemented a variant of A2C and Impala in which the value network had full observation of the game. We observed a significant decrease in training speed and no clear performance improvement, therefore, decided not to use it further.

4. We tried to implement Contrastive Predictive Coding [35] as well, however, we faced implementation issues and decided to prioritize our efforts on other ideas.

5. We implemented weight sharing on various levels; the whole population, roles (e.g. preferences in the Alleathoric Harvest substrate), and personas (described below).

While vanilla self-play worked well on some substrates, in most cases it lacked stability and generalizability, we addressed these problems with the following approaches.

1. We defined personas for each role, e.g., prosocial and anti-social ones whose reward signals are positive or negative concerning the population's average reward, respectively. Agents randomly chose a persona at the beginning of each episode to follow and optimize.

2. Reward sharing between agents either for the whole population, roles, or personas.

3. We tuned the reward signal for each specific substrate to overcome their respective challenges. Details follow in the next section.

4. We divided training into phases in increasing difficulty and introduced new social aspects in each stage. Mainly used for Territory Rooms and details are described in the respective section.

5. We introduced pre-trained bots from the evaluation scenarios. It did not help significantly while slowing down our training due to implementation difficulties. We did not use this for the final solution.

6. We experimented with sampling different scenarios for each episode during training in an attempt to enhance generalization performance. However, the training process became unstable because of divergent reward ranges, and normalizing them did not resolve the issue. We opted to abandon this approach due to time constraints.

**Substrate Specifics**

**Alleathoric Harvest:** The substrate faces the challenge of effectively deciding between planting the personal favorite berry type and planting the globally dominant berry type since it ripens faster. Given that agents possess only partial observations of the environment, identifying the dominant berry type is difficult, and reaching a consensus on the type of berry to plant among agents is equally challenging.

To address this challenge, we implemented a custom reward function. Agents are incentivized to plant a predefined berry (red), which we manually selected. Additionally, agents receive instant rewards for converting other berries to the predefined type. On the other hand, there is a penalty for switching from the predefined berry to other types to prevent the accumulation of rewards through constant back-and-forth changes. Due to time constraints, we were unable to implement a dynamic predefined berry, one that rewards based on the actual global dominating berry.

All agents sharing the same role are trained with a shared-weight policy to enhance sample efficiency. We also fine-tuned rewards for zapping (to encourage more aggressive behaviors) and being zapped. In the end, we achieved optimal performance by not rewarding zapping and penalizing being zapped with a -2 reward.

**Prisoner's Dilemma:** We believe the ideal strategy for this substrate is cooperating with the members from the focal population while defecting the background population. However, it is intrinsically difficult to distinguish focal agents from background agents only based on image observations. Therefore, we started by training pure cooperating strategy in this challenge, because the cooperating strategy generally provides a higher mean focal outcome in scenarios compared with the purely defective strategy. To do this, we modified the outcome matrix to give a reward of 3 for mutual cooperation and zero for any other outcome. All agents were trained with a shared policy for higher sample efficiency.

Another challenge for this substrate is the lack of resources. The default rewards encouraged taking as many resources as possible. In this case, some agents will obtain all the resources, while others fail to gather any resources to interact with others. We addressed this problem by designing penalties for consuming more than one resource per agent. This further improved the performance according to our evaluations. Due to time limits, we were not able to investigate learning the aforementioned ideal strategy to behave depending on the prediction of the "identity" of the other agents, which could be an interesting future work.

**Clean Up:** The main challenge in this scenario is the sparse reward feedback. The agent can only receive rewards by eating apples, however, if they do not know how to clean the river efficiently, there will be no apple growing and as a result, no reward feedback. Therefore, we utilize reward shaping to encourage the agents to clean dirty blocks. More concretely, we provide additional rewards for agents, such that, on the one hand, it is proportional to the number of polluted blocks they cleaned to encourage more efficient cleaning, and on the other hand, it is also lower than the reward of tasting apples so they will not interpret that cleaning river is their fundamental goal.

Besides, each agent takes the average reward of all agents as the reward in training, and in this way, they are encouraged to have prosocial behavior. Moreover, we also leverage the symmetricity of agents by sharing the policies for all the agents, and we observe it results in more sample-efficient learning.

**Territory Rooms:** Our biggest challenge in this substrate was to encourage exploration of the agents, i.e., breaking walls and entering other rooms while learning how to defend themselves. Our final approach was based on a three-step learning procedure as follows.

1. Teach a single agent with others being idle. A larger additional reward was given to claim the resources the first time and a smaller one to reclaim them. The agent learned to explore the whole state space, break walls, and claim resources while ignoring the other agents.

2. Two agents learning concurrently with 7 others being idle. Both agents were initialized from the one in the previous step and they share weights. We provided additional positive rewards for initially claiming resources and zapping others while negative rewards for reclaiming resources, using the beams, and being zapped. We aimed to maintain the incentive to explore while engaging in conflicts and learning how to defend themselves. Penalizing zapping led to fewer resources being destroyed.

3. In the last step, we used the same structure as before but with 3 or 4 agents. We observed marginal improvements only on top of the previous results.

A further observation about this substrate was that PopArt and Impala helped for stability but a smaller learning rate was better for PopArt compared to the values reported in the whitepaper. With the initial learning rate, PopArt scaled the value function too quickly and agents fell into a local optima where they did not leave their own room.

