# OpenReview forum: "Melting Pot Contest: Charting the Future of Generalized Cooperative Intelligence"
_NeurIPS.cc/2024/Datasets_and_Benchmarks_Track — NeurIPS 2024 Track Datasets and Benchmarks Poster_

### Official Review · Reviewer_mWg1 · 2024-07-22
**Good Contest Paper but Lack of Insights for Future Research**

**Rating:** 6
**Confidence:** 4
**Clarity:** Yes.

**Review:**

This paper clarifies the gap in the generalization of cooperative intelligence, while it does not present some details about the contest, and has some typos in presentation.

**Strengths:**

This paper analyzes the submitted solutions in the contest and provides the results and effectiveness of these solutions.

**Additional Feedback:**

If the details of background agents are further discussed, I will consider improving my rating.

**Correctness:**

This paper uses constructed background agents to evaluate the submitted solutions, while the background agents are not introduced and discussed. The lack of these details affects the soundness of evaluation results.

**Documentation:**

Yes.

**Ethics:**

No.

**Limitations:**

Yes.

**Opportunities For Improvement:**

For presentations:
1. there is one typo in Line 261.
2. The citations in 4.3 are confused; e.g., IMPALA is first introduced without citation.

For content:
This paper lacks details on the construction of background agents and analyses of those background agents.

**Relation To Prior Work:**

Yes.

**Summary And Contributions:**

The submitted paper details the Melting Pot Contest 2023, which focuses on evaluating cooperative intelligence of AI agents in multi-agent environments. The analysis highlights performance on various substrates, robustness of scoring metrics, and in-group coordination behaviors. Key contributions include a detailed performance analysis using Inter-Quartile Mean (IQM) to validate rankings, and a capability profile assessment using XGBoost regression models. Additionally, the study explores agent behaviors in the "Clean Up" scenario, where focal agents' interactions influence background agents' actions, demonstrating coordination without antisocial outcomes. The paper also describes the design of measurement layouts for evaluating agent capabilities, emphasizing predictive power and model interpretability.

---

> ### Author Rebuttal · Authors · 2024-08-20
>
> We thank the reviewer for their time and efforts in reviewing our manuscript. Below we address the reviewer’s concerns and request to raise further questions if something is still not clear.
>
> > Presentation issues such as typo and citations
>
> Thank you for catching them. We will fix them and proofread the entire manuscript again to avoid such occurrences in the revised version of the paper.
>
> > Constructions of Background Agents
>
> Thank you for bringing this concern to our attention. In the initial draft, we omitted certain details because the background agents used in the contest were either identical to or constructed using the same processes outlined in the original Melting Pot papers [1,2]. However, we acknowledge the reviewer's point that it would be beneficial for readers to have these details included within this paper. With that in mind, we will first outline the general procedure for constructing the background agents, and then describe how they were used in the contest. We will add a dedicated section in the revised version of the paper to cover these aspects comprehensively.
> The background population consists of reinforcement learning (RL) agents, referred to as "bots" to differentiate them from the focal population's agents. The creation of the background population involved three key steps: (1) specification, (2) training, and (3) quality control.
>
> **Specification:** The designer typically starts with an idea of what they want the final bot’s behavior to look like. Since substrate events provide privileged information about other agents and the environment, we can often specify reward functions that easily induce the desired behavior. This task is much simpler than the challenge faced by focal agents, who must learn from pixels and final rewards alone. However, when a single reward function is insufficient to capture the desired behavior, we employ techniques inspired by hierarchical reinforcement learning [3,4,5], such as reward shaping [6] and the "option keyboard" [7].
>
> To generate complex behaviors, we first train bots using different environment events as reward signals, similar to the approach used in Horde [3]. These behaviors are then combined using simple Python scripts, allowing us to express complex behaviors in an "if this event, then run that behavior" manner. This approach, which we call the "puppet" method, uses the same basic neural network structures (ConvNet, MLP, LSTM) as other agents but introduces a hierarchical policy structure.
>
> For example, in the Clean Up task, we designed a bot that cleans only when other players are cleaning. The architecture is inspired by Feudal Networks [8], but with key differences. We represent goals as a one-hot vector $g$, which is embedded into a continuous representation $e(g)$. This embedding $e$ is then provided as an additional input to the LSTM. The network outputs several policies $\pi_z(a∣x)$, and the final policy is a mixture $\pi(a∣x)=\sum_z\alpha(e)\pi_z(a∣x)$, where the mixture coefficients $\alpha(e)=\textbf{SoftMax}(e)$ are learned from the embedding. Notably, instead of directly associating policies with goals, we allow the embedding to learn these associations through experience.
>
> **Training:** To train the puppet to follow goals, we train it in the respective environment with goals switching at random intervals and rewarding the agent for following them. The thing to keep in mind is that the bots must generalize to the focal population. To this end, we chose at least some bots—typically not used in the final scenario—that are likely to develop behaviors resembling that of the focal agent at test time. For instance, in Running With Scissors in the Matrix, we train rock, paper, and scissors specialist bots alongside “free” bots that experience the true substrate reward function.
>
> **Quality control:** Bot quality control is done by running 10–30 episodes where candidate bots interact with other fixed bots. These other bots are typically a mixture of familiar and unfamiliar bots (that trained together or separately). We verify that agents trained to optimize for a certain event, indeed do. We reject agents that fail to do so.
>
>
> > How were these background populations used in the contest?
>
> As outlined in the paper, the contest was divided into three phases: (i) Development Phase, (ii) Generalization Phase, and (iii) Evaluation Phase. During the Development Phase, participants were evaluated on 22 different background populations (corresponding to 22 scenarios) across four substrates. These populations were originally created for the foundational papers [1,2], and the results in those papers were based on these background populations. Both the populations and the puppet codes used to create them were publicly available and accessible to participants.
>
> For the Generalization and Evaluation Phases, we created a new set of 51 background populations across the same four substrates. Participants were evaluated on these held-out sets, without access to the background populations or their specifications. In the Generalization Phase, we sampled 12 of these populations (3 per substrate) and provided scores to the participants, allowing them to adjust their models based solely on these scores. During the Evaluation Phase, participants were tested against all 51 background populations without any opportunity to adjust their models, ensuring a robust generalization test.
> The specifications and puppet codes for both the original 22 and the newly constructed 51 background populations used in this contest are now publicly available [here](https://github.com/google-deepmind/meltingpot/blob/main/meltingpot/configs/scenarios/__init__.py).
>
> We hope that this helps to alleviate the main concern raised by the reviewer but please let us know if there still remains any outstanding concern that would prevent the reviewer from revisiting their score.

---

> > ### Author Response · Authors · 2024-08-20
> > **References for the above comment**
> >
> > [1] Melting Pot 2.0, Agapious et. al. 2023
> >
> > [2] Scalable Evaluation of Multi-Agent Reinforcement Learning with Melting Pot, Leibo et. al. 2021
> >
> > [3] Horde: A scalable real-time architecture for learning knowledge from unsupervised sensorimotor interaction, Sutton et al., 2011
> >
> > [4] Universal value function approximators, Schaul et al., 2015;
> >
> > [5] Between mdps and semi-mdps: A framework for temporal abstraction in reinforcement learning, Sutton et al., 1999
> >
> > [6] Reinforcement learning: An introduction, Sutton & Barto, 2018
> >
> > [7] The option keyboard: Combining skills in reinforcement learning Barreto et al., 2019
> >
> > [8] Feudal networks for hierarchical reinforcement learning, Vezhnevets et al., 2017

---

> > > ### Comment · Reviewer_mWg1 · 2024-08-22
> > > **Clear Clarification**
> > >
> > > The authors clearly explain the construction of the background populations. It will be beneficial to include this part in the main text.
> > >
> > > I raise the score from 4 to 6.

---

### Official Review · Reviewer_wn2e · 2024-07-24
**A helpful contribution on a topic that is highly relevant for the benchmark track**

**Rating:** 7
**Confidence:** 3
**Correctness:** Yes.
**Clarity:** Yes. It's very well written.

**Review:**

Details below. The paper is extremely well written and was a pleasure to read. Though I highlight opportunities for minor improvements, I consider this a strong submission overall.

**Strengths:**

S1) Highly relevant for the NeurIPS benchmark track, this paper exemplifies how insights from a large-scale competition can further work on cooperative AI benchmarks.

S2) The paper is well-scoped, providing a thorough yet succinct overview of the competition design, a useful analysis of the results, and a helpful overview of top strategies.

S3) The mixed quantitative and qualitative analyses are well-designed and yield useful insights.

**Additional Feedback:**

.

**Documentation:**

Yes.

**Ethics:**

Nothing to flag.

**Limitations:**

As noted above, the paper would benefit from an expanded limitations discussion.

**Opportunities For Improvement:**

Minor opportunities for improvement:

* The paper would benefit from a more explicit discussion of the limitations of both the melting pot benchmark and the contest.

* It would also benefit from a more sustained discussion of areas of future work: including with similar contests, the melting pot benchmark, and cooperative intelligence benchmarks more generally.

**Relation To Prior Work:**

Yes.

**Summary And Contributions:**

The paper shares findings from the melting pot contest, a challenge in which participants from 100+ teams submitted potential solutions to the melting pot benchmark (http://arxiv.org/abs/2211.13746).  It also offers insights into the benchmark and the problem of measuring performance on tasks that involve “cooperation among interacting agents” and “promot[ing] (differential) progress on the cooperative intelligence of AI systems.” The core sections of the paper discuss the design of the Melting Pot contest (Section 2), the results of the contest (Section 3), and strategies from the top submissions (Section 4).

---

> ### Author Rebuttal · Authors · 2024-08-20
>
> We thank the reviewer for their time and effort in reviewing our manuscript. We are pleased to see the reviewer's appreciation of the scope and analysis presented in our work. We are also grateful for the insightful feedback regarding the limitations and future directions of our research. In Section 5, we have already touched upon several points in this regard, including a reference to a follow-up contest planned for NeurIPS 2024. However, based on this feedback, we are happy to include a more detailed discussion on these aspects in the revised version of the manuscript.

---

> > ### Comment · Reviewer_wn2e · 2024-08-26
> > **Thank you!**
> >
> > Thank you for engaging with the feedback, and for your valuable paper. I hope the paper makes it through the review process.

---

### Official Review · Reviewer_e1NQ · 2024-07-25
**Benchmarking multi-agent learning to cooperate**

**Rating:** 7
**Confidence:** 4
**Correctness:** the claims are still correct, at leas…
**Clarity:** the paper is clear and well written.

**Review:**

Benchmarking cooperative skills of multiple agents is important. In this case the idea leveraging the melting pot is novel.
pros
- The paper is well written and clear.
- The melting pot benchmark and its evaluation metric is original
cons
- The setting and detail procedure of (whether) learning is involved during the contest is not completely described.

**Strengths:**

- Benchmarking cooperative skills of multiple agents is important. In this case the idea leveraging the melting pot is novel.
- The paper is well written and clear.
- The melting pot benchmark and its evaluation metric is original

**Additional Feedback:**

some questions and suggestions have been given above in the limitations and room for improvements above.

**Documentation:**

the URL to access the contest is provided. Only the detail about the learning curve on the spot as mentioned is not given.

**Ethics:**

I don't think there's an ethical concern here. However, since this paper is based on a contest with so many participants involved that have been conducted in the past or in prior work, I think it should be better also to include a documentation or some kind of formal approval related to the ethical matter regarding the contest.

**Limitations:**

to address the limitations as mentioned, the paper can be added with learning curves of each methods or algorithms as parts of the performance measures.
The clarity of the domain environment can still be improved by indicating which one is the substrate, which is the focal agents, etc. maybe directly in the figures.

**Opportunities For Improvement:**

Adaptation towards cooperative behavior requires rounds of interaction for the focal agents to adapt with unknown agents in a new environment. The paper does not describe whether the focal agents are still allowed to learn during the contest. If they can or were facilitated to learn on the spot, how long or how many iterations for performance to converge is given to the participants in the contest? perhaps the paper can be added with learning curves of each methods or algorithms as parts of the performance measures.

The clarity of the domain environment can still be improved by indicating which one is the substrate, which is the focal agents, etc. maybe directly in the figures.

**Relation To Prior Work:**

the prior work is adequately discussed.

**Summary And Contributions:**

The paper present a new benchmark as a contest called Meting Pot for a multiagent team to learn cooperative behavior from other (novel) agents.

---

> ### Author Rebuttal · Authors · 2024-08-20
>
> We thank the reviewer for their time and effort in reviewing our manuscript. We are delighted to see the positive remarks of the reviewer on the originality and importance of this work.
>
> Below we clarify the confusion surrounding the training setup for the focal agents and details on substrates, scenarios and agents.
>
> > Substrate, Scenarios, Focal Agents and Background population
>
> As explained in Section 2 of the paper, the term "substrate" refers to the physical aspects of the environment, including its spatial layout, object placement, and the rules of physics. Essentially, a substrate can be considered equivalent to any reinforcement learning (RL) environment. In this contest, we used four substrates: CleanUP, Allelopathic Harvest, Prisoners’ Dilemma in the Matrix, and Territory: Rooms.
>
> Focal agents are those trained on the substrates and submitted by the participants. Training focal agents on a substrate is akin to the standard process of training agents in any RL environment.
>
> A scenario is defined as the combination of a substrate and a background population. The "background population" refers to the agents, other than the focal agents, that inhabit the environment. While focal agents experience the substrate during training, they do not interact with the background population.
>
> The background population is only used for evaluation and agents in the background population are trained by organizers and not the participants.
>
> > Training and evaluation
>
> We provide detailed information on the training and evaluation pipeline in Section 2 and Appendix B. Briefly, the process is as follows:
> Participants trained their focal agents on a substrate without any background population agents present. Background agents were introduced only during the evaluation phase. However, as discussed later, participants did have access to a few background populations during training. Participants were free to use any training approach without any restrictions, allowing them to consider the properties of the background populations available to them for local evaluation, though this carried the risk of overfitting.
>
> Once participants were satisfied with the performance of their trained agents (the focal population), they could submit their agents at any time. They were required to submit a single focal population capable of performing across all four substrates.
>
> For evaluation, a set of background agents and a set of focal agents (from the participants' submissions) were sampled, and the evaluation protocol described in Section 2.2 (lines 96-100) and Appendix B was followed.
>
> Participants had two signals to guide adjustments to their models: the local evaluation score (based on scenarios already accessible to them) and the leaderboard score, which was computed and reported by the organizers by evaluating the submitted focal population against the background population.
>
> The contest was divided into three phases: (i) Development Phase, (ii) Generalization Phase, and (iii) Evaluation Phase.
> During the Development Phase, participants train focal agents on substrates and submit at any time they want (with a restriction of 2 submissions per day due to compute constraints). They were evaluated on 22 different background populations (corresponding to 22 scenarios) across four substrates. These populations were publicly available and accessible to participants.
>
> For the Generalization and Evaluation Phases, we created a new set of 51 background populations across the same four substrates. Participants were evaluated on these held-out sets, without access to the background populations or their specifications. In the Generalization Phase, we sampled 12 of these populations (3 per substrate) and provided scores to the participants, allowing them to still adjust their models based solely on these scores (different from the development phase where they also had access to the background agents, not just scores from organizers). During the Evaluation Phase, participants were tested against all 51 background populations without any opportunity to adjust their models, ensuring a robust zero-shot generalization test, the key objective of this contest.
>
> > Training curves
>
> Since participants were allowed to train their focal agents as many times as needed during the development and generalization phases, and were only required to submit the final trained agents, we do not have access to their local training curves. However, we refer the reviewer to [1, 2] for detailed training curves and comparisons for some of the algorithms benchmarked in those papers.
>
> [1] Melting Pot 2.0, Agapious et. al. 2023
>
> [2] Scalable Evaluation of Multi-Agent Reinforcement Learning with Melting Pot, Leibo et. al. 2021

---

### Decision · Program_Chairs · 2024-09-26

**Decision:**

Accept (Poster)

**Comment:**

The overall motivation of this paper is to advance research towards the development of cooperatively intelligent agents. To this end, the authors present findings from the Melting Pot 2023 challenge. The paper outlines the design of the challenge and summarizes the results before discussing strategies of the winning submissions. The authors also discuss some of the lessons learned.

I believe the paper is very relevant to the D&B track and exemplifies how challenges and benchmarks can bring researchers together (over 600 participants across 100+ teams) and drive research as a community. I also concur with the authors that the paper is well written. Given the overall quality of the work and the discussions with the reviewers, I recommend accepting this paper to the program.

I recommend that the authors include parts of the rebuttal into the final version of the paper, especially the clarification to reviewer mWg1 and the expanded discussion of limitations suggested by reviewer wn2e.

One additional aspect that the reviewers did not mention, but might be relevant to making the paper stronger, is expanding the discussion on how choices of operationalizing social welfare might affect the findings or future challenges. The authors do acknowledge alternative measures of social welfare beyond utilitarian welfare in footnote 6, but from the results I wonder if leveraging some of the other measures of welfare might penalize specific undesirable behaviors or capture different perspectives towards a more robust benchmark.